# Sound Can Help Us See More Clearly

**DOI:** 10.3390/s22020599

**Published:** 2022-01-13

**Authors:** Yongsheng Li, Tengfei Tu, Hua Zhang, Jishuai Li, Zhengping Jin, Qiaoyan Wen

**Affiliations:** State Key Laboratory of Networking and Switching Technology, Beijing University of Posts and Telecommunications, Beijing 100876, China; lee_yongsheng@bupt.edu.cn (Y.L.); tutengfei.kevin@bupt.edu.cn (T.T.); sky_lee1990@bupt.edu.cn (J.L.); zhpjin@bupt.edu.cn (Z.J.); wqy@bupt.edu.cn (Q.W.)

**Keywords:** sound texture, two-stream network, computer vision

## Abstract

In the field of video action classification, existing network frameworks often only use video frames as input. When the object involved in the action does not appear in a prominent position in the video frame, the network cannot accurately classify it. We introduce a new neural network structure that uses sound to assist in processing such tasks. The original sound wave is converted into sound texture as the input of the network. Furthermore, in order to use the rich modal information (images and sound) in the video, we designed and used a two-stream frame. In this work, we assume that sound data can be used to solve motion recognition tasks. To demonstrate this, we designed a neural network based on sound texture to perform video action classification tasks. Then, we fuse this network with a deep neural network that uses continuous video frames to construct a two-stream network, which is called A-IN. Finally, in the kinetics dataset, we use our proposed A-IN to compare with the image-only network. The experimental results show that the recognition accuracy of the two-stream neural network model with uesed sound data features is increased by 7.6% compared with the network using video frames. This proves that the rational use of the rich information in the video can improve the classification effect.

## 1. Introduction

The sheer volume of video data nowadays demands robust video classification techniques that can effectively recognize human actions and complex events for applications such as video search, summarization, or intelligent surveillance. However, due to the complexity of the information provided by the video, including classification conflicts caused by different viewing conditions and noise content unrelated to the video theme, this is a particularly challenging problem. At the same time, when the proportion of the objects interacting in the action is too small, and there is no prominent position displayed, it is difficult distinguish the action category effectively using only the image information in the video. We can use other information in the video file to assist the classification, such as sound (as shown in Figure 1). Sound can convey important information about the world around us, and sound can be used as a supplement to visual information in some cases. The sound in the video originates from the interaction between objects. Therefore specific audio can be the main discriminator for certain actions (such as “washing”) and objects in the action. Due to these correlations, we believe that the sound information that occurs in synchronization with the visual signal in the video can also provide rich training features, which can be used to train the video action classification model.

The research on the use of audio data in video mainly focuses on the following aspects: audio–visual representation learning [1,2,3,4,5,6], sound-source localization [7,8,9], audio–visual source separation [5,10,11,12,13,14], visual question answering [15], and generating sounds from video [16,17,18,19,20]. However, in the task of video action classification, the audio information in these videos is rarely used. Audio signal may provide information that that is largely orthogonal to that available in images alone—information about semantics, events, and mechanics are all readily available from sound [21]. Moreover, the usual method for the use of sound is to use the original waveform data of the sound as the input of the neural network, or simply perform a two-dimensional conversion. These processing methods do not consider the characteristics and correlations of different frequencies in the sounds emitted by the objects participating in the action interaction, that is, the sound texture.

The deep neural networks that can learn features automatically from raw data have demonstrated strong performance in various domains. In particular, the convolutional neural networks (ConvNets) are very successful in image analysis tasks such as object detection [22], object recognition [23,24], and image segmentation [25]. However, for video classification, most deep network based approaches [26,27,28,29] demonstrated worse or similar results to the hand-engineered features [30]. This is largely due to the high complexity of the video data. This requires researchers to effectively use data such as visual images and auditory audio tracks in video files. One of the key factors of the action classification system is the design and use of good features. For this reason, we chose the sound texture in the video as the feature input of the network to assist the classification of the action.

Based on the correlation between the image and sound in the video, we propose a method that uses the rich features provided by the video frame and audio at the same time. For this reason, we designed a two-stream neural network that uses the two-modal information of the image and the sound in the video to process the video action classification task. The following are the contributions of our work:We propose a neural network structure for solving video action recognition, which uses the sound texture in the video as input. Experiments have proven that the trained model can achieve an effect similar to a network using images;We designed a two-stream neural network structure, which integrates the spatio-temporal and audio cues in the video. The two branch networks describe the video from different angles, and the results are fused in a linearly weighted manner. Experiments have shown that the recognition accuracy of this neural network is higher than that of a single branch.

The rest of this article is organized as follows. Section 2 reviews and discusses related work. Section 3 describes the proposed sound processing method and the two-way network framework in detail. Section 4 shows and compares the experimental results, and Section 5 shows our conclusions.

## 2. Related Works

**Action Recognition**: Action recognition in video has been extensively studied in recent decades. Research has transitioned from the initial use of hand-designed local spatiotemporal features [30,31,32] to mid-level descriptors [33,34,35], and recently transitioned to end-to-end learning deep video representations. Various deep neural networks have been proposed to simulate the spatio-temporal information in the video [36,37,38,39,40]. Recent work includes capturing a long-term temporal structure via recurrent networks [41,42] or ranking functions [43], pooling across space and/or time [44,45], modeling hierarchical or spatiotemporal information in videos [46,47], and building long-term temporal relations [48,49].

The current network design usually uses a deep neural network with a 3D convolution kernel to extract features in video frames. The problem that this brings is that, as the depth of the neural network increases, the parameters that need to be trained are also increasing rapidly, and resource limitations have been called the bottleneck of deep learning. When there is a multi-stream network, the difficulty of training will be further increased. In order to alleviate this situation, we study the use of background sounds in the video as features to train a lightweight network model to assist in improving the classification accuracy of the model.

**Audio–Visual Learning**: Over the last three years, significant attention has been paid in computer vision to an underutilised and readily available source of information existing in video (the audio stream). These fall into one of five categories: audio–visual representation learning [1,2,3,4,5,6], sound-source localization [7,8,9], audio–visual source separation [5,10,11,12,13,14], visual question answering [15], and generating sounds from video [16,17,18,19,20]. Different from all the above work, the main content of our work is to effectively use audio to assist in the recognition of video actions.

**Processing Sound**: For the use of audio in video, most of the existing work focuses on using sound as a continuous input, making it suitable for recurrent neural networks such as LSTM, or applying a short-time Fourier transform to convert a one-dimensional sound track into a two-dimensional image (that is, a spectrogram), where the horizontal axis and the vertical axis are the time scale and the frequency scale, respectively. Through this conversion, it can be applied to the common convolutional neural network structure in the image field.

The above two kinds of processing methods ignore the correlation between different frequencies in the background sound, which is very important for action classification. We use the brain’s processing method of sound for reference, and let the original sound wave pass through multiple different filters to simulate the filtering process of sound by the cochlea. Then, the filtering results are mathematically counted to obtain the sound texture.

**Multi-Stream Network**: Video contains abundant multimodal information, so a multi-stream framework is proposed to fully utilize the rich multimodal information in videos [28,50,51]. Considering the consumption of computing resources, the existing work mostly focuses on the research of two-stream neural network. The two branches of a two-stream network often adopt a similar network structure, and merge at the end of the network to realize the function of action classification. Common network inputs are single frame image, a set of continuous images, a set of extracted images, and optical flow.

This inadvertently ignores the information provided by the sound in the video. Moreover, with the increase in network depth, the number of training parameters of image related depth neural network will increase greatly, which also improves the training difficulty to a certain extent. Therefore, we design a network structure for sound texture. It is integrated with the network using video frames through a two stream network structure. Through the two-way structure, the full use of multi-modal information in video is realized. Compared with the network using video frames, only a small amount of training parameters are needed to significantly improve the accuracy of classification.

## 3. Materials and Methods

Our goal is to use sound to accurately and efficiently perform video action recognition. First, we define the sound processing method, that is, how to generate the sound texture; then, we introduce how to use the sound texture data to construct a neural network that solves the problem of video action recognition. Finally, we built a two-stream network to make full use of the information provided by the sound and images in the video.

### 3.1. Sound Processing

For sound data, a natural question is how should our model perform feature processing on the sound in the video. The first method that comes to mind is to directly convert the sound waves in the video into an atlas. This transformation can convert a one-dimensional sound track into a two-dimensional image. However, this is potentially suboptimal; as the sound source in the video is complex, this processing method cannot express the relationship between different sources, which is a crucial condition for action recognition.

In order to effectively use audio information, we use the sound texture model of McDermott and Simoncelli [52], which assumes that sound is stationary within a temporal window. This method imitates the process of the human brain processing sound. The sound waveform passes through different filters within a fixed period of time to simulate the working principle of sound filtering by the cochlea, and performs further statistical calculations on the data of each channel to obtain the final result. The process of sound texture is shown in Figure 2.

In this article, we retain the structure of the original processing method and adjust the details. For the sound extracted from the video file, the detailed process is as follows:(1)We extract the original audio data within a fixed time from the video, that is, a one-dimensional audio waveform.(2)We filter the audio waveform with a bank of 20 bandpass filters intended to mimic human cochlear frequency selectivity.(3)We then take the Hilbert envelope of each channel.(4)We raise each sample of the envelope to the 0.3 power (to mimic cochlear amplitude compression) and resample the compressed envelope to 400 Hz. We compute time-averaged statistics of these subband envelopes: we compute the mean and standard deviation of each frequency channel, the mean squared response of each of a bank of modulation filters applied to each channel, and the Pearson correlation between pairs of channels.(5)For the subband envelopes, we use a bank of 10 band-pass filters with center frequencies ranging from 0.5 to 200 Hz, equally spaced on a logarithmic scale.(6)For the filtering results in step 5, we calculate the marginal moments of each sub-channel, the intra-channel correlation C1, and the correlation C2 of the corresponding channel.(7)We divide the calculation results in steps 4 and 6 (i.e., marginal moments, correlation) by the dimensional value as the sound texture, and finally we obtain a 320-dimensional vector.

In the Algorithm 1 description, steps 1, 2, 3, 4, 9, and 10 show the cochlear processing method to process the original audio file. The remaining steps show the operation process of sound texture data. Among them, SoundWave represents the audio data obtained after the audio file extracted from the video file is processed by the fast Fourier transform. Channeln represents the data of the Hilbert envelope formed after the audio data is filtered by the nth band pass filter. Channelmn represents the result obtained after Channeln passes through a set of 10 bandpass filters.
**Algorithm 1** Micro Attention branch Algorithm**Require:**SoundWave: Sound waveform data extracted from video;**Ensure:**SoundTexture1:**for** each n∈[1,20] **do**2:    Channeln = Hilbert(BandPassFiltern(SoundWave));3:    Channeln = Channeln**0.3;4:    Channeln = Compress(Channeln);5:    Marginaln = Marginal(Channeln);6:    MarginalData = Append(MarginalData, Marginaln)7:    CorrelationCn = Correlation(Channeln,Channel20−n)8:    CorrelationDate = Append(CorrelationDate,CorrelationCn)9:    **for** each m∈[1,10] **do**10:        Channelnm = BandPassFilterm(Channeln);11:        Marginalnm = Marginal(Channelnm);12:        MarginalData = Append(MarginalData, Marginalnm)13:        CorrelationC1m = Correlation(Channelnm,Channeln(10−m))14:        CorrelationC2m = Correlation(Channelnm,Channel(20−n)m)15:        CorrelationDate = Append(CorrelationDate,CorrelationC1m,CorrelationC2m)16:    **end for**17:**end for**18:SoundTexture = Append(MarginalData,CorrelationDate)19:**return**SoundTexture

### 3.2. Audio Network

**Audio Network**: Aiming at the data structure of the generated sound texture, we designed a neural network to use the sound texture. As the input of the sound texture is a 1*N matrix (N represents the number of statistical results in the sound texture, the value in this article is 320), this structure is not suitable for the convolutional neural network architecture of deep neural networks. This is as the purpose of the basic element convolution kernel in the convolutional network is to extract the features of adjacent elements in the matrix (N*M), and the sound texture feature is only applicable to the 1*1 convolution kernel. In addition, the data contained in the sound texture feature is the result of statistical calculations within a period of time, and there is no temporal relationship between the data, which makes the data unsuitable for sequence-related recurrent neural networks. Therefore, we finally chose the basic supervised multi-layer feedforward neural network framework to handle the multi-classification task of actions.

We have designed a neural network constructed with five hidden layers and an output layer (called AN) to process action classification tasks in videos. The network structure is shown in Figure 3. The network can conveniently use one-dimensional vectors as input, that is, 1*N sound texture matrix. Each element in the sound texture data obtains the value of each node of the hidden layer through a weighting operation. The result of each hidden layer will be processed by the ReLU activation function and batch normalization. The number of nodes in the five hidden layers are 128, 128, 64, 64, and 32, and the structure is shown in Table 1. In order to reduce the number of parameters that need to be trained in the network structure and avoid over-fitting, the random drop rate of hidden layer nodes in the network is set to 50%. The network can train the action classification model by using sound texture to optimize the task of action classification.

### 3.3. Network Architecture

Video carries a wealth of multi-modal information, usually showing the movement and interaction of objects over time in a specific scene, accompanied by human voices or background sounds. Therefore, video data can be naturally decomposed into spatiotemporal information streams and audio streams. The spatio-temporal information can be represented by consecutive frames in the video. The sound in the video provides important clues for action classification, and these clues are usually complementary to visual counterparts. In order to make full use of the multi-modal information provided by the video, inspired by the two-stream network, we propose a two-stream network structure that uses flames and sound, called A-IN. The structure consists of two branches that use different structure data. One is a deep neural network using video frames to extract data features of spatio-temporal information streams; the other is a sound texture neural network we proposed to extract data features of audio streams, as shown in Figure 4.

The working mode of the two-stream neural network is: the two branches of the neural network use their respective data inputs to independently extract features and give their respective prediction results. At the end of the two-steam neural network, the results of the two branches will be fused by linear weighting as the final prediction result.

In our proposed A-IN, on the branch corresponding to the space–time dimension, we adopted the deep network architecture I3D [37], which has excellent effects on the action recognition task. It can effectively recognize certain video semantics that have clear and discriminative appearance characteristics. The network uses a set of continuous frames of video as the input of the model, and the continuous video frames of a video segment can be regarded as a matrix of [*T*, *C*, *H*, *W*] dimensions. The network is composed of basic modules of the same structure stacked sequentially. With this design, as the network depth increases, high-level features in the video frame can be extracted. The basic module (as shown in Figure 5) contains four 3D convolution kernel branches, which can be used to extract data features in different directions of interest in the spatiotemporal information stream. The network finally outputs the probability of each action classification, and uses this as the prediction result.

In the branch network of audio stream, we adopt the network structure proposed in Section 3.2. We separately extract the synchronized track data in the video and process it according to Section 3.1 to obtain the 1*320 dimension sound texture feature data, which is used as the input of the sound texture network branch. Finally, the branch also outputs the probability of each action classification, and uses this as the prediction result.

As the two branches of the network have different network data structures, the data features extracted by different branches cannot be merged in the mid-term. We choose to perform linear weighted fusion on the output of the two branches at the end of the network structure. The final calculation of the prediction result of the network is as Formula (1). Through such a network design, we can effectively use the image information in the video and the auxiliary information of the sound in the video.
(1)prediction=w∗I3D(Frames)+(1−w)∗AN(SoundTexture)

Formula (1) shows the calculation process of the two-stream neural network to obtain the final prediction result. The I3D branch of the network that uses video frames as input and the AN branch that uses sound texture operate independently, and the final prediction result is obtained by linear weighting the calculation results of the two. The input of the I3D network is Frames, and its dimensions are [*N*, *T*, *C*, *H*, *W*], where *N* represents the number of videos in each batch, *T* represents the number of frames of each video, *C* represents the number of channels, *H* represents the height of the frame, and *W* represents the width of each frame. The input of the Audio Network is Sound Texture, whose dimensions are [1, *N*], where *N* represents the number of various mathematical statistics in the sound texture.

Our proposed framework can integrate audio and images, and comprehensively model video data. As mentioned above, such a framework consists of two separately trained deep networks. Although being feasible to jointly train the entire framework, it is complicated and computationally demanding. In addition, training multiple deep networks separately makes the approach more flexible, where a component may be replaced without the need of re-training the entire framework. This means that we can use other discriminative network frameworks to replace the existing network to obtain better performance. As described in the next section, through experiments on the proposed audio and image two-stream network, it is proven that the framework is effective in video classification tasks.

## 4. Experiments

In this section, we trained and tested the results of the above network model on the Kinetics dataset. We designed experiments to verify the following three problems: (a) Can networks using sound texture handle video action classification tasks? (b) Does sound texture improve classification performance compared with sound spectrogram? (c) Does the two-stream network with added sound input improve the classification accuracy compared with the original I3D?

### 4.1. Experimental Setup

Dataset: The experimental training and test data used the open source video data set Kinetics [53], and we randomly selected seven categories (exercising with an exercise ball, parasailing, washing dishes, pull ups, cleaning shoes, folding paper, and pumping fist). Moreover, we removed videos without background sound from the data set to filter out a new dataset. We followd the suggested experimental protocol and reported mean accuracy over the three training and test splits.

ComparedMethods: In order to verify the effectiveness of our proposed network, we compare the following networks:

I3D: A neural network which only uses video frames and is the state of art in the field of motion recognition. We processed the videos in the dataset through the CV tool, and extracted 80 frames of images with a resolution of 224 * 224 from each video as the input of the network. If the entire video was less than 80 frames, we looped the entire video to make up 80 frames.

AudioNetwork: We propose a network using sound texture to verify whether sound can be used to deal with action classification tasks. We used the network structure using sound texture proposed in Section 3.1. We extracted 3.75 s of audio track data from the video, and formed a 320-dimensional vector as the input of the network according to the processing method in Section 3.1.

ConvNet: We employed ConvNet [54] as the baseline. The network uses sound spectrogram to verify the effectiveness of manually designed sound texture. We first applied the Short-Time Fourier Transformation to convert the 1-D soundtrack into a 2-D image (a spectrogram) with the horizontal axis and vertical axis being time-scale and frequency-scale, respectively.

**Two-Stream**: Following to this scheme [37], we used two identical I3D to construct a two-way network. It has achieved excellent results in action classification tasks at this stage. The two branches of the network use video frames and optical flow, respectively, as inputs. The processing method of video frames is shown in I3D. We computed optical flow with a TV-L1 algorithm [55].

The above-mentioned network is independently trained from scratch under the same resource conditions. The training environment of these networks is python version V3.6, the machine learning library used is Tensorflow, and its version is V1.5. Training on videos used standard SGD with momentum set to 0.9 in all cases. Models were trained using a similar learning rate. At the same time, in the two-stream neural network (A-IN), the weighted operation value *w* of the output results of the two branches is set to 0.5.

### 4.2. Results and Discussions

Table 2 reports the results. Comparing the first two rows of Table 2, presents interesting results. We found that the classification accuracy of AN that uses sound texture as the input is very close to that of I3D, only 2.6% lower. Thus, we design a network using sound texture as input, which can be used to deal with action classification tasks. Moreover, we reviewed the incorrectly classified video clips and found that some video clips replaced the original sound with background music.

Then, we compared the data in rows 2 and 3 of Table 2. We found that the accuracy of the baseline that directly uses the sound spectrogram data as input is 6.2% lower than that of AN. In the field of motion recognition, the use of manually designed sound texture features will have better results than the use of original sound spectrogram. This is as the sound texture takes into account the differences of starting sounds of different actions and the correlation between different audio in the sound. Furthermore, features such as sound texture involve complex statistical calculations, which are difficult to extract through deep neural networks, so using artificially designed data features will achieve unexpected results.

Focusing on the last row of Table 2, we found that the two-stream network (A-IN) using both sound and video frames achieved the best results. Compared with the I3D network, its accuracy is improved by 7.6%. This shows that the reasonable use of multi-mode information of video can effectively improve the effect of action classification. Moreover, sound can indeed be used as a supplement to video frames to assist in solving action classification tasks. We found that the accuracy of the A-IN network is 2.3% higher than that of network using video frame and optical flow. However, the two stream network uses two I3D networks, therefore its parameters are twice higher than A-IN. This greatly increases the resource consumption in the process of training and verification.

Figure 6 further shows the performance of different models on each category in the dataset. We can clearly see that the two-stream neural network brings consistent and more obvious improvements to all classes. That is to say, the reasonable use of multi-modal information in video can effectively improve the effect of artificial intelligence in action recognition tasks. We found that the accuracy of AN is low for such ’pull up’ actions, which is due to the noise in the video of such actions, which has an impact on the network model.

From Table 3, we notice that the total parameters of AN which need to be trained are only 70 K, while the amount of parameters that need to be trained for the I3D network is as high as 12.3 M. We can therefore draw the conclusion that our proposed two-stream neural network using sound and image, compared with I3D, only needs to increase a small number of parameters to improve the effect of the action classification task. Due to the huge difference in the parameters of the two branch networks, under the same hardware resources and time conditions, we can further improve the effect of the AN network through more iterations. Additionally, as the sound texture requires a variety of statistical calculations on the original sound, the processing time for input data is higher than the processing time for directly extracting video frames.

## 5. Conclusions

The fact that videos contain multi-modal information requires neural networks not only to understand static visual information, but also to explore movement and auditory cues. The background sound in the video has rich data information and can be used as auxiliary data for video action recognition. In this work, in order to make full use of the data features in the video to improve the classification accuracy, we introduce the background sound in the video to assist the action classification. We transform the original sound into sound texture by imitating the human processing of sound. Then, we construct an audio network to extract the features of sound texture. Finally, we combine the network I3D using video frames to construct a two-stream neural network called A-IN. Compared with the I3D network using only image features, the network significantly improves the accuracy of network action classification with a small amount of new parameters. In theory, the audio network in this paper can be fused with other deep neural networks through two-stream network structure, and the effect needs to be further verified.

## Figures and Tables

**Figure 1 sensors-22-00599-f001:**
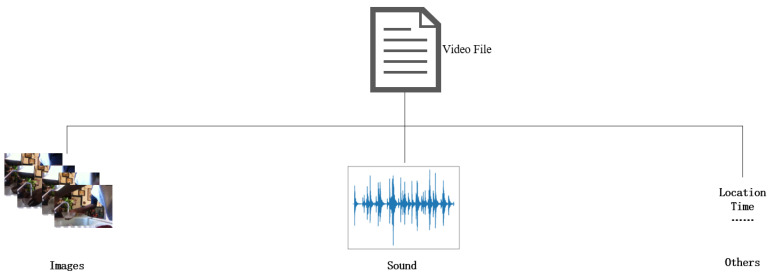
The multimode data in the video file.

**Figure 2 sensors-22-00599-f002:**
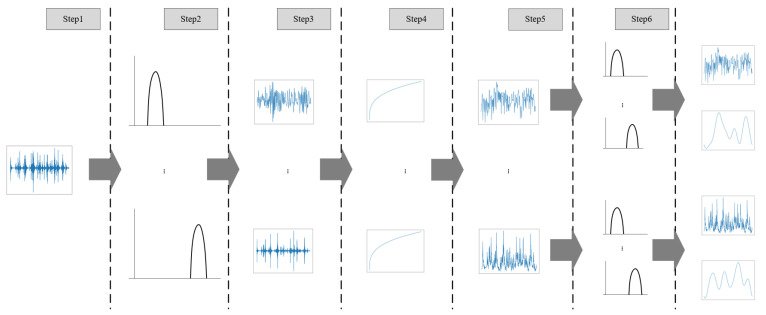
The specific process of obtaining sound texture from original audio.

**Figure 3 sensors-22-00599-f003:**
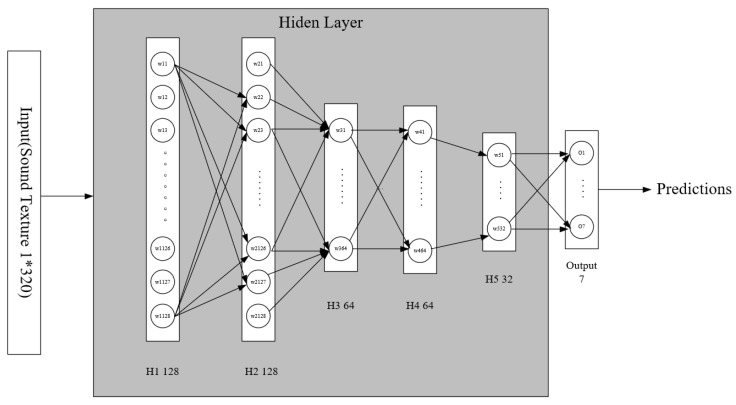
The structure of the neural network we built, which consists of five hidden layers and input and output layers.

**Figure 4 sensors-22-00599-f004:**
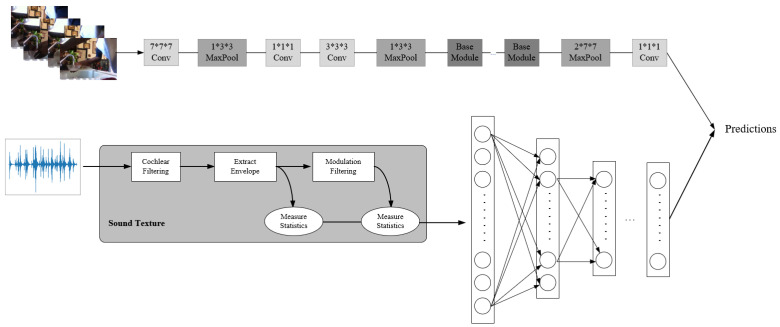
The simplified structure of the two-stream neural network (A-IN).

**Figure 5 sensors-22-00599-f005:**
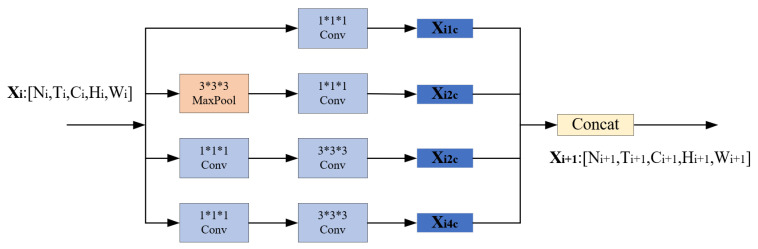
The basic module structure of I3D.

**Figure 6 sensors-22-00599-f006:**
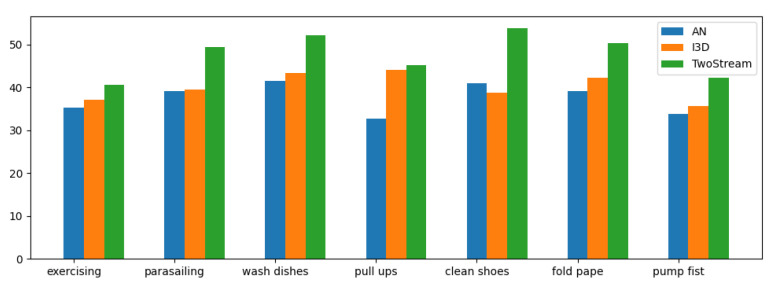
The prediction accuracy of different networks under different action types.

**Table 1 sensors-22-00599-t001:** Structure of the audio network.

Network Layer	Nodes	Drop	Activation	Normalization
hidden layer1	128	0.5	ReLU	True
hidden layer2	128	0.5	ReLU	True
hidden layer3	64	0.5	ReLU	True
hidden layer4	64	0.5	ReLU	True
hidden layer5	32	0.5	ReLU	True
output layer	7	0		

**Table 2 sensors-22-00599-t002:** Structure of the network.

Network	Input	Top1	Top2
I3D	Frames	40.1%	58.8%
AN	Sound Texture	37.5%	52.7%
ConvNet	Spectrogram	31.3%	48.9%
Two-Stream	Frames + Optical Flow	45.4%	62.1%
A-IN	Frames + Sound Texture	47.7%	63.2%

**Table 3 sensors-22-00599-t003:** Resources and time consumed by the network.

Network	Params	Operation Time	Prediction Time
I3D	12.3 M	0.27 s	2.2 s
AN	70 K	2.5 s	0.1 s

## Data Availability

Not applicable.

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
