# Peer review of "Sound Can Help Us See More Clearly"

_sensors, 2022, doi:10.3390/s22020599_

Round 1

Reviewer 1 Report

The authors shared an interesting approach for video action classification. The manuscript overall is good, but I would recommend several sections to be more detailed:

  1. Please add a paragraph in the introduction section where the structure of the manuscript is described.
  2. The manuscript has a relative extended related work section, but I would recommend the authors to be better highlight what is the difference between the proposed scheme and existing ones.
  3. Please share information about the learning phase, results of the fine-tuning and so on.
  4. Section 4 is too sublime, and I did not clearly understand what exactly was tested. Is this method and benchmark for this kind of activities? And I would like to understand how this proposes solution compare with some of the solutions presented in section Related work.
  5. Please reconsider section 5, it is too succinct and lacks the actual ideas that were novel for this manuscript.

Reviewer 2 Report

This paper presents a method that uses sound in video classification tasks. The motivation is convincing and the results are interesting, which are worthwhile to publish. I only have some minor comments that the author should consider in the final version. 

Figure 2 is not clear.

Algorithm 1 Micro Attention branch Algorithm, the authors should write the summary the proposed method in a more mathematical form.

The first letter of the title and subtitle should be capitalized. For example, 3.3. network architecture.

In Figure 4, it is hard to read the content of the word.

The writing should be improved someplace. Please proofread the manuscript to improve the quality. There are also many errors in the reference.

Round 2

Reviewer 1 Report

I recommend accepting the manuscript in this form. Thank you for considering the remarks.